# The Encapsulation of Citicoline within Solid Lipid Nanoparticles Enhances Its Capability to Counteract the 6-Hydroxydopamine-Induced Cytotoxicity in Human Neuroblastoma SH-SY5Y Cells

**DOI:** 10.3390/pharmaceutics14091827

**Published:** 2022-08-30

**Authors:** Andrea Margari, Anna Grazia Monteduro, Silvia Rizzato, Loredana Capobianco, Alessio Crestini, Roberto Rivabene, Paola Piscopo, Mara D’Onofrio, Valeria Manzini, Giuseppe Trapani, Alessandra Quarta, Giuseppe Maruccio, Carmelo Ventra, Luigi Lieto, Adriana Trapani

**Affiliations:** 1Omnics Research Group, Department of Mathematics and Physics “Ennio De Giorgi”, University of Salento and INFN Sezione di Lecce, Via per Monteroni, 73100 Lecce, Italy; 2CNR-NANOTEC Institute of Nanotechnology, Via Monteroni, 73100 Lecce, Italy; 3Department of Biological and Environmental Sciences and Technologies, University of Salento, 73100 Lecce, Italy; 4Department of Neuroscience, Istituto Superiore di Sanità, Viale Regina Elena, 00161 Rome, Italy; 5European Brain Research Institute (EBRI) “Rita Levi-Montalcini”, Viale Regina Elena, 00161 Rome, Italy; 6Department of Pharmacy-Drug Sciences, University of Bari “Aldo Moro”, Via Orabona, 70125 Bari, Italy; 7Esseti Farmaceutici, Via Cavalli di Bronzo, 39-46, San Giorgio a Cremano, 80046 Naples, Italy

**Keywords:** citicoline, solid lipid nanoparticles, 6-hydroxydopamine, SH-SY5Y dopaminergic cells

## Abstract

(1) Backgrond: Considering the positive effects of citicoline (CIT) in the management of some neurodegenerative diseases, the aim of this work was to develop CIT-Loaded Solid Lipid Nanoparticles (CIT-SLNs) for enhancing the therapeutic use of CIT in parkinsonian syndrome; (2) Methods: CIT-SLNs were prepared by the melt homogenization method using the self-emulsifying lipid Gelucire^®^ 50/13 as lipid matrix. Solid-state features on CIT-SLNs were obtained with FT-IR, thermal analysis (DSC) and X-ray powder diffraction (XRPD) studies. (3) Results: CIT-SLNs showed a mean diameter of 201 nm, −2.20 mV as zeta potential and a high percentage of entrapped CIT. DSC and XRPD analyses evidenced a greater amorphous state of CIT in CIT-SLNs. On confocal microscopy, fluorescent SLNs replacing unlabeled CIT-SLNs released the dye selectively in the cytoplasm. Biological evaluation showed that pre-treatment of SH-SY5Y dopaminergic cells with CIT-SLNs (50 µM) before the addition of 40 µM 6-hydroxydopamine (6-OHDA) to mimic Parkinson’s disease’s degenerative pathways counteracts the cytotoxic effects induced by the neurotoxin, increasing cell viability with the consistent maintenance of both nuclear and cell morphology. In contrast, pre-treatment with CIT 50 and 60 µM or plain SLNs for 2 h followed by 6-OHDA (40 µM) did not significantly influence cell viability. (4) Conclusions: These data suggest an enhanced protection exerted by CIT-SLNs with respect to free CIT and prompt further investigation of possible molecular mechanisms that underlie this difference.

## 1. Introduction

In the last decade, it has been increasingly evident that the blood–brain barrier (BBB) represents the most critical concern for the treatment of age-related neurodegenerative diseases (NDs) because it significantly limits the access to the brain of neuroprotective drugs in therapeutic amounts [1]. Moreover, recently, the mechanisms leading to a progressive loss of neuronal function up to neuronal death have been better elucidated for many NDs. They include oxidative stress [2], neuro-inflammation [3], apoptosis [4] and protein aggregation [5]. Thus, for instance, in the case of Parkinson’s disease (PD), which represents the second most frequent ND in the developed countries yet is still of unclear etiology, new therapeutic targets are available for the treatment and reduction of the progression towards dopaminergic midbrain neuron degeneration. In particular, these new therapeutic targets include α-synuclein (α-syn) aggregation and phosphorylation, the proteosome lysosome approach, mitochondrial disruptions and oxidative stress as well as neuro-inflammation and the glial environment [6]. To date, there are no drugs capable of modifying the course of PD, only interventions for the symptoms. The main one is L-DOPA with some dopamine agonists such as catechol-O-methyl transferase inhibitors, monoamine oxidase-B (MAO-B) inhibitors and non-dopamine drugs (such as anticholinergics and amantadine). In this framework, the need for other therapies hampering disease progression is extremely relevant.

Citicoline (CIT, Cytidine 5’-Diphosphocholine or cytidine diphosphate choline, Figure 1) is a choline donor used as an intermediary in phosphatidylcholine biosynthesis. After oral or parenteral administration, CIT provides two principal components, cytidine and choline. Hence, CIT is a nutritional supplement and source of choline and cytidine with potential neuroprotective and nootropic activity (PubChem CID 13804). CIT has been used as a therapeutic agent in combination with L-Dopa in parkinsonian syndrome [7,8,9]. Even though the rationale behind this last use still remains to be fully clarified, CIT positively increases the effects of treatment with L-Dopa, improving motor symptoms and cognitive impairment in patients with PD [10,11,12,13]. Due to its hydrophilic features (calculated Log P-4, computed by XLog P3 3.0, PubChem release 7 May 2021), CIT is unable to cross the BBB.

Among the available strategies for bypassing the BBB, including receptor-mediated transcytosis [14] and the simple chemical modification of the therapeutic agent leading to prodrugs and codrugs [14,15,16,17,18], in recent years, particular interest has been focused on the intranasal administration of therapeutic agents encapsulated in nanocarriers. This last approach offers direct and noninvasive access to the brain of the neuroprotective drug, exploiting the anatomical connections between the respiratory region of the nasal cavity and olfactory bulb and the trigeminal nerve [19,20]. Compared with the oral administration route, characterized both by patient compliance and support of pharmaceutical industry [21,22], the intranasal administration can provide faster brain delivery, avoid gastrointestinal and hepatic metabolism and reduce systemic side effects [23]. Moreover, intranasal administration may also enable drug absorption by the lymphatic system followed by successive transport into the systemic circulation [24]. Therefore, nose-to-brain delivery can be a valuable strategy for ND treatment.

Drug encapsulation in nanocarriers provides several advantages not only in terms of the protection of the encapsulated drug from possible degradation in the nasal cavity, which thus enables higher bioavailability and permeation, but also in terms of BBB crossing. Indeed, these neuroprotective encapsulated agents can efficiently overcome the BBB, exploiting physiological transport mechanisms such as adsorptive-mediated and receptor-mediated transcytosis [25,26,27]. Moreover, several natural or synthetic biomaterials have been identified as inhibiting the P-glycoprotein (P-gp) efflux pump, which is the most represented on BBB. This evidence opens up possibilities of increasing the bioavailability of Pgp-substrate neuroactive drugs after nano-encapsulations in biomaterials [28].

In this context, it is noteworthy that, comparing lipid-based nanocarriers with other types of nanostructured systems for brain delivery shows that lipid-based nanosystems have several advantages over the polymer ones [29,30]. These last may show toxicity issues for the presence of organic solvent residues and difficulty with large-scale production [23]. Based on the above, lipid-based nanocarriers have great potential not only for drug administration by airways [31] but also in the treatment of neurological diseases [29].

Among the available lipid based nanocarriers, solid lipid nanoparticles (SLNs) showed several favorable features such as safety and stability. They can be prepared on a large scale and administered following several delivery routes. In particular, SLNs are extensively used in addition to cyclodextrins and polymeric micelles [32,33] to improve the formulation of hydrophobic drugs, although hydrophilic drugs can also be encapsulated in these nanocarriers [34]. SLNs have been used as delivery systems to cross the BBB when intravenously injected and therefore represent an opportunity for ND treatment [34,35]. Moreover, they can also be intranasally administered as it is, formulated in combination with mucoadhesive hydrogels or coated with mucoadhesive polymers to improve the management of PD [36,37]. Indeed, the use of these mucoadhesive biomaterials, extensively employed for biomedical applications [38,39], limits the mucociliary clearance, enhancing the retention time in the nasal mucosa.

In the present work, we encapsulated CIT in SLNs for potential intranasal administration and evaluated the effects of these nanocarriers on 6-OHDA-treated SH-SY5Y dopaminergic cells with the aim to shed light even on the mentioned therapeutic CIT use in parkinsonian syndrome. It should be noted that while CIT encapsulation in liposomes or in niosomes was previously reported [31,40], to the best of our knowledge, CIT-loaded SLNs (CIT-SLNs) have not yet been described in the literature. They were prepared, for the first time, in the present work by the melt homogenization method using the self-emulsifying lipid Gelucire^®^ 50/13 as the lipid matrix. We have previously demonstrated that the mentioned preparative method applied to a self-emulsifying lipid such as Gelucire^®^ 50/13 also allows for efficiently loading hydrophilic substances in the resulting SLNs. Hence, this method offers an alternative to double emulsification (W/O/W) and is advantageous since it is also organic solvent free [41,42,43]. Because Gelucire^®^ 50/13 is composed of PEG esters (stearoyl polyoxyl-32 glycerides), a small glyceride fraction and free PEG chains, the corresponding SLNs are PEGylated SLNs. The results obtained in these investigations are reported and discussed below.

## 2. Materials and Methods

### 2.1. Materials

Gelucire^®^ 50/13 was kindly supplied by Gattefossè (Milan, Italy). Citicoline sodium salt (CIT) was a gift from Esseti Farmaceutici s.r.l. (Pomezia, Italy). Polysorbate 85 (Tween^®^ 85), acetic acid (AcOH), KBr, porcine stomach mucin (type II, sialic acid ~1%), Fluorescein isothiocyanate (FITC) and PBS were purchased from Sigma Aldrich (Milan, Italy). 6-Hydroxydopamine hydrobromide (6-OHDA) was provided by LKT Laboratories Inc. (St. Paul, MN, USA). Throughout this work, double-distilled water was used. All other chemicals were of reagent grade.

Human neuroblastoma cell line SH-SY5Y was cultivated in DMEM low-glucose medium with stable glutamine (Euroclone, Milan, Italy) supplemented with 10% heat-inactivated (*v*/*v*) fetal bovine serum (Euroclone, Milan, Italy), penicillin (100 IU/mL), and streptomycin (100 mg/mL) in a humidified atmosphere at 37 °C with 5% CO_2_. For the experiments, the cells were seeded at a density of 3.1 × 10^4^/cm^2^ on 96-well round plates and were grown for 1 day before toxic treatments (6-OHDA 40 µM). Free CIT or CIT-SLNs (50 or 60 µM referred to CIT) were administered 2 h before 6-OHDA and incubated for 24 h. Phase-contrast images of cultures were carried out using a Fluovert Fu inverted microscope (Wild Leitz GMBH, Wetzlar, Germany) and digitized using a CCD camera (MoticamPro 252A, Wetzlar, Germany).

### 2.2. Quantitative Determination of Citicoline

The quantitative determination of CIT in Gelucire^®^ 50/13-based SLNs was carried out with spectrophotometric analysis using a calibration curve obtained by dissolving CIT in double distilled water (concentration range 6–0.6 mg/mL, R^2^ > 0.999). The measurements were performed at the wavelength of 280 nm using a Perkin-Elmer Lambda Bio 20 UV-Vis spectrophotometer (Perkin-Elmer, Milan, Italy). This successful quantitative determination of CIT by UV spectrophotometry was carried out according to Sachan et al. [44], who demonstrated the high selectivity, linearity, accuracy and precision of the method. Moreover, in our hands, no substance other than CIT had any absorption at the selected UV wavelength.

### 2.3. Preparation of Citicoline Loaded Gelucire^®^ 50/13 Based-SLNs (CIT-SLNs) and Fluorescent FITC-SLNs

CIT-SLNs were prepared according to the melt emulsification method [41,42,43]. Briefly, 60 mg of Gelucire^®^ 50/13 were melted at 70 °C, and in a separate vial, 1.37 mL of a diluted AcOH solution (0.01%, *w*/*v*) containing the surfactant (Tween 85^®^, 60 mg) was heated at 70 °C. In the AcOH solution, 10 mg of CIT were poured prior to the addition of the resulting mixture to the melted lipid at 70 °C. Then, an emulsion was obtained by homogenization at 12,300 rpm for 3 min, with an Ultra-Turrax model T25 apparatus (Janke and Kunkel, IKA^®^-Werke GmbH & Co., Staufen, Germany). Finally, the nanosuspension was cooled at room temperature, and the resulting SLN were centrifuged (Eppendorf 5415D, Hamburg, Germany) at 13,200× *g*, 45 min and employed for subsequent studies; the supernatant was discarded. Lyophilized samples of CIT-SLNs were obtained after freeze drying over 72 h (Lio Pascal 5P, Milan, Italy) without the use of any cryoprotectant.

Fluorescent FITC-SLNs, used for confocal microscopy studies, were prepared as reported in [42]. Briefly, SLNs were prepared as described above except in the aqueous phase, CIT was replaced with 6 mg of FITC.

### 2.4. Physicochemical Characterization of CIT-SLNs Prepared

Particle size and polydispersity index (PDI) of SLNs were determined with a ZetasizerNanoZS (ZEN 3600, Malvern, UK) apparatus in photon correlation spectroscopy (PCS) mode. Each sample was diluted 1:1 (*v*:*v*) with double-distilled water. Laser Doppler anemometry (ZetasizerNanoZS, ZEN 3600, Malvern, UK) was used to determine the zeta-potential values after the dilution of the sample 1:20 (*v*:*v*) in the presence of KCl (1 mM, pH 7). The particle size, PDI and zeta potential values were each measured in triplicate, and the results are shown as the mean ± SD.

The morphology of the CIT-SLNs was evaluated through transmission electron microscopy (TEM). The suspension in water of CIT-SLNs (25 µL) was dropped onto carbon-coated copper grids and stained with 1% (*w*/*v*) phosphotungstic acid solution (pH 7.4) for 1 min. Then, the grid was rinsed with deionized water and left to dry under air overnight. Low-magnification TEM images of the SLN were recorded on a JEOL Jem1011 microscope (Tokyo, Japan) operating at an accelerating voltage of 100 kV.

The amount of CIT loaded in the SLNs, i.e., the encapsulation efficiency of CIT (E.E. CIT%) was calculated as follows:E.E. = Total CIT − CIT in the SLN supernatant/Total CIT × 100
where the Total CIT was the total initial amount of drug employed in the SLN preparation, and CIT in the SLN supernatant was the CIT amount determined after the centrifugation and separation of the supernatant. This last amount was determined by spectrophotometric analysis as reported in Section 2.2.

### 2.5. Solid State Studies

#### 2.5.1. Fourier Transform Infrared (FT-IR) Spectroscopy

FT-IR spectra of pure CIT, freeze-dried samples of plain SLNs and CIT-SLNs were recorded in KBr discs using 2–5 mg of each of these samples and a Perkin Elmer 1600 FT-IR spectrometer (Perkin Elmer, Milan, Italy) operating in the range 4000–400 cm^−1^ with a resolution of 1 cm^−1^.

#### 2.5.2. Differential Scanning Calorimetry (DSC)

DSC thermograms of pure CIT, freeze-dried samples of plain SLNs and CIT-SLNs were obtained with a Mettler Toledo DSC 822e instrument (Mettler Toledo, Milan, Italy), using 40 μL capacity aluminum crucibles and an empty pan as a reference. Aliquots of about 5 mg of each sample were placed in the pans, which were heated from 25 °C to 280 °C at a rate of 5 °C/min, under a nitrogen flow of 50 cm^3^/min. Indium powder was used as a standard for temperature calibration. Each experiment was carried out in triplicate.

#### 2.5.3. X-ray Powder Diffraction (XRPD)

The crystal structures of CIT, freeze-dried samples of plain SLNs and CIT SLNs were investigated by acquiring the X-ray diffraction patterns using a X’Pert PRO (PANalytical, Malvern, UK) system. The data were collected at room temperature in the 2θ range of 5–40° with a step size of 0.02°.

### 2.6. Physical Stability of CIT-SLNs on Storage

The physical stability of the CIT-SLNs was evaluated by measuring their size after incubation at two temperatures, i.e., at 4° C up to 3 months and at 37 °C up to 24 h. The particle size was measured at different time intervals following the procedure reported in Section 2.4. Each experiment was carried out in triplicate.

### 2.7. Biological Evaluations

#### 2.7.1. Cell Viability Assay

Cell viability was determined using the Cell Counting Kit 8 assay (Enzo Life Sciences, Euroclone, Milan, Italy) according to the manufacturer’s instructions. This test is based on the mitochondrial enzyme-dependent reaction of a water-soluble tetrazolium salt for quantifying the number of live cells by producing an orange formazan dye upon bio-reduction in the presence of an electron carrier. Briefly, cell cultures were grown under the different experimental conditions for 24 h before replacing the medium with 100 µL of a 10% (*v*/*v*) mixed solution of the WST-8/CCK8 probe in fresh DMEM. Cell viability after 4 h incubation was determined with a spectrophotometric measure of absorbance at 450 nm using an ELISA microplate reader (Tecan, Sunrise, Männedorf Switzerland). The data were normalized to those of the 6-OHDA to measure relative cell viability with respect to CIT and SLNs resuspended after freeze-drying.

#### 2.7.2. Uptake Assay with FITC-SLNs

SH-SY5Y cells were incubated for 24 h with FITC-loaded SLNs (50 and 100 µg/mL) or plain SLNs as auto-fluorescence control. Thereafter, cells were washed with PBS, fixed for 15 min with 4% paraformaldehyde and rinsed in PBS. To visualize the nuclei, material was mounted with Seebright mounting medium with diamidino-2-phenylindole (DAPI, Enzo Life Sciences, Farmingdale, NY, USA). The slides were examined with a Zeiss LSM 980 confocal laser-scanning microscope (Zeiss, Oberkochen, Germany).

#### 2.7.3. DAPI Staining

Cell nucleus appearance and the relative condensation and/or fragmentation were determined with DAPI staining. SH-SY5Y cells were cultured on coverslips for 24 h, pre-treated with free CIT or CIT-SLNs (50 µM) and grown in the presence of 6-OHDA (40 µM) for an additional 24h of incubation. After this time, the glass culture dishes were washed with PBS and fixed in paraformaldehyde (4%) at room temperature for 20 min. Afterward, cells were washed three times in PBS and mounted in Seebright Mounting Medium with DAPI (Enzo Life Sciences, Farmingdale, NY, USA) and observed under a confocal inverted microscope (IX73 Olympus equipped with a V2 confocal spinning disk module, Crisel Instruments, Rome, Italy) with a 20× and a 40× oil immersion objective illuminated with ultraviolet light and detected through a blue/cyan filter. The acquisition and processing of images were obtained using MetaMorph software 7.8.1. (Molecular Devices, San Josè, CA, USA).

### 2.8. Statistics

Statistical analyses were carried out in Prism v. 5, GraphPad Software Inc. (San Diego, CA, USA). Data were expressed as mean ± standard deviation. Multiple comparisons were based on repeated ANOVA measurements together with Bonferroni’s multiple comparison test, and differences were considered significant when *p* < 0.05.

## 3. Results

### 3.1. Formulation and Characterization of CIT-SLNs

The melt homogenization method, which we already employed in previous studies [41,42,43], allowed for the preparation of CIT-SLNs by adding to the melted solid lipid Gelucire^®^ 50/13 at 70 °C, an aqueous solution containing AcOH, the surfactant (Tween^®^ 85) and CIT. After cooling at room temperature, the resulting emulsion obtained by the homogenization of the whole mixture gave rise to the desired nanocarriers.

In Table 1 the main physicochemical properties of CIT-SLNs are shown. In detail, the CIT-SLN particles were significantly larger than the plain SLNs used as control with a broad size distribution as shown in Figure 1b, in agreement with the observed Polydispersity indices (PDIs) (0.45 ± 0.08). As for the zeta potential values of these nanocarriers, CIT-SLNs showed a slightly more positive value (−2.2 mV vs. −9.2 mV, respectively, exhibited by the plain SLNs). The E.E. CIT% for CIT-SLNs was good enough, confirming once again the satisfactory application of the preparative method to a hydrophilic compound as CIT. As shown in Figure 1d–f, to visualize the morphology of CIT-SLNs, TEM microscopy was adopted, evidencing spherically shaped core-shell particles, often joined among them, giving rise to elongated spherical and irregularly shaped particles.

### 3.2. Solid State Studies

To gain information on the solid-state features of the prepared CIT-SLNs, spectroscopic (FT-IR), thermal analysis (DSC) and X-ray powder diffraction (XRPD) studies were performed, and the corresponding results are shown in Figure 2 and Figure 3.

The FT-IR spectra of the CIT-SLNs showed absorption bands occurring both in pure CIT and in the plain SLNs (Figure 2). Thus, the peak at 1737 cm^−1^, detectable in the FT-IR spectrum of CIT-SLNs attributable to the ester carbonyl function of Gelucire^®^ 50/13, was observed also in plain SLNs, while the absorption band at 1656 cm^−1^ of FT-IR spectrum of CIT-SLNs was also present in the FT-IR spectrum of pure CIT and for the peak at 1472 cm^−1^ of CIT-SLNs, which also occurs in the spectrum of pure CIT, although slightly moved to 1487 cm^−1^. Overall, it may be deduced that the FT-IR spectrum of CIT-SLNs is reminiscent of pure CIT and plain SLNs FT-IR spectra.

The DSC thermogram of pure CIT (Figure 2) in our hands showed two endothermic peaks at 130 °C and 150 °C as well as an exothermic peak at about 260 °C. Taking into account that the CIT melting is reported to occur in the range 259–268 °C [45], presumably the peak at about 260 °C should be due to melting with exothermic decomposition. In addition, at the present time, it is difficult to rationalize the presence of two endothermic peaks at 130 °C and 150 °C before the melting of the drug, and more detailed investigation in this regard is required. The DSC profile of CIT-SLNs showed a broad endothermic peak at 49 °C and small signals at 55 °C, both attributable to the solid-lipid Gelucire^®^ 50/13 [46,47]. In addition, the very small endothermic peak at about 122 °C and the exothermic one at 250 °C were also present. Hence, in the DSC profile of CIT-SLNs are present peaks due to pure CIT and plain SLNs (Figure 2). Even the XRPD results confirm what was observed on FT-IR spectroscopy and in DSC analysis (Figure 3): the XRPD diffractogram of pure CIT indicates its crystalline nature, showing the characteristic peaks of the pure drug as reported in literature [48]. The diffractogram of SLN without CIT shows the most intense peaks at 19° and 23° [49,50]. CIT-SLNs show a similar XRD pattern, suggesting a similar crystal structure to that of plain SLNs in addition to a most reduced crystallinity of the drug [51].

### 3.3. Physical Stability on Storage of CIT-SLNs

Figure 4a,b show the particle size variation observed on storage at 4 °C and 37 °C within 3 months and 24 h, respectively, for CIT-SLNs. In the former case, the particles increased their mean diameters up to almost double their original value after 1 month of storage, while in the latter case, the increase of about 10% was noted after 1 day of storage.

### 3.4. Biological Evaluations

#### 3.4.1. SLNs Are Able to Deliver a Fluorescent Dye inside the SH-SY5Y Cells

The release of the tracer FITC from SLNs to the cell cultures is shown in Figure 5a,b. Confocal images show that 24 h after the administration of the SLNs loaded with 50 or 100 μg/mL FITC, the dye is diffusely localized into the cytoplasm but not in the nuclei. Cells incubated with plain SLNs in Figure 5c confirm the absence of auto-fluorescence.

#### 3.4.2. Pre-Treatment with CIT Loaded SLNs Rescues Cell Viability in SH-SY5Y Cells Exposed to the Neurotoxic 6-OHDA

Cell viability of SH-SY5Y cells, assessed in the CCK-8 assay after 6-OHDA treatment, is shown in Figure 6. Treatment with 6-OHDA (40 µM) for 24 h produced a significant decrease of 17.1% (*p* < 0.001; *n* = 5) in cell survival. A pre-treatment with 50 and 60 µM CIT for 2 h followed by 6-OHDA (40 µM) after 24 h did not significantly influence cell viability with respect to the driving toxic effect exerted by 6-OHDA. Conversely, the 50 µM concentration of CIT when embedded in SLNs increased significantly the cell survival to 14% (CIT-SLNs 50 µM vs. 6-OHDA treated cells, *p* < 0.01, *n* = 5) in comparison with 6-OHDA treated cells and CIT-SLNs 60 µM although showing an increase in viability (Δ viability = 9.5%) respect to the 6-OHDA treated cells it did not reach significance. The effect of plain SLNs was unable to modify cell viability with respect to 6-OHDA (40 µM) treatment. Moreover, the comparison between CIT SLNs 50 and CIT 50 µM (Δ viability = 11.7%, *p* < 0.05, *n* = 5) and 60 µM (Δ viability = 12.8%, *p* < 0.01, *n* = 5) shows that SLNs produce a slight, yet significant, increase in viability after exposure to 6-OHDA.

#### 3.4.3. Cell Morphology of Cell Culture

Representative phase-contrast micrographs show a consistent effect on cell morphology, due to the presence of cell retraction and swelling with the appearance of cellular debris, in cell cultures treated with 6-OHDA (Figure 7B,L) with respect to untreated cell cultures (Figure 7A,I). No obvious difference was found in the cell morphology of cultures pre-treated with free CIT (Figure 7C,M) and CIT-SLNs (Figure 7D,N) in comparison with cell cultures exclusively exposed to 6-OHDA. In cultures pre-treated with CIT-SLNs and plain SLNs (data not shown), a quote of the amorphous material is due to nanoparticle aggregates precipitating on the cell surface. As shown in Figure 7F,P, the DNA damage induced by 6-OHDA was also detected on DAPI staining. Compared with the control group (Figure 7E,O), cells exposed to the toxic treatment show nuclear condensation and DNA fragmentation (arrows) as markers of cell apoptosis. Pre-exposure to free CIT (Figure 7G,Q) was unable to modify the toxic effects of 6-OHDA, as shown in micrographs that resemble the nuclear fragmentation present in cell cultures treated with 6-OHDA. It is noteworthy that the CIT-SLN treatment (Figure 7H,R) reduces the effect of 6-OHDA-induced DNA fragmentation and chromatin condensation, showing a lightening nuclear condense rate.

## 4. Discussion

The present work was aimed at shedding light on the possible mechanism(s) accounting for the therapeutic use of CIT in parkinsonian syndrome since there are studies suggesting that CDP–choline supplements increase dopamine receptor densities [52]. For this purpose, CIT-loaded SLNs (i.e., CIT-SLNs) were prepared, and in view of a nasal administration to bypass the BBB, the effects of these nanocarriers on 6-OHDA-treated SH-SY5Y dopaminergic cells were investigated. The preparative method for CIT-SLNs was the melt homogenization method using a self-emulsifying solid lipid matrix as Gelucire^®^ 50/13. Since CIT is a hydrophilic substance, it was added to water phase in significant amounts due to chemical affinity for this phase, rather than the co-melting approach with the lipid Gelucire^®^ 50/13 which was adopted for hydrophobic drug substances loading [47]. Precisely, the high amount of CIT herein used results greater than that occurs by adding the same substance to the lipid phase. We have widely shown in a series of previous papers that this method allows for preparing not only lipophilic but also hydrophilic-loaded SLNs in good EE% [41,42,43,46]. We demonstrated that this preparative method for hydrophilic substances using a self-emulsifying lipid can be an alternative to the standard double emulsification (W1/O/W2) method with the relevant advantage of being organic solvent free.

As shown in Table 1, the nanocarriers were characterized by mean diameters of about 200 nm as well as by a broad distribution as confirmed by high PDIs and as evidenced by PCS (Figure 1). The essentially neutral zeta potential value observed for CIT-SLNs (i.e., −2.2 ± 0.2 mV) compared with that more negative of plain SLNs (i.e., −9.2 ± 0.7 mV) suggests the lower colloidal stability of the former particles than the latter ones. Moreover, the surface charge of CIT-SLNs could be interpreted in the context of the model we proposed for these Gelucire^®^ 50/13 based SLNs. We proposed that these PEGylated SLNs consist of a hydrophilic shell of polyoxyethylene chains of Gelucire^®^ 50/13 and cosurfactant (Tween^®^ 85) as well as of an internal lipid core constituted by the stearoyl moieties of the solid lipid [42]. This indicates that a hydrophilic substance as CIT could be adsorbed on the particle surface or entrapped in the hydrophilic shell as well as encapsulated in the lipid core as a nanoemulsion [42]. In the case of CIT, whose diphosphocholine moiety is positively charged, adsorption on the particle surface and/or entrapment into hydrophilic shell may occur by the interaction between the oxygen atom of oxyethylene groups and the diphosphocholine moiety. The increase of diameter size from 141 nm (plain SLNs) to 201 nm (CIT-SLNs) may be due to the fact that a hydrophilic substance as CIT could be adsorbed even on the particle surface of the outer hydrophilic shell of the particle, enlarging the corresponding diameter. In this regard, we previously observed increases in diameter size from plain SLNs to drug-loaded SLNs [41]. The essentially neutral zeta potential value of CIT-SLNs suggests that in these nanocarriers, the positive charge density on the surface is higher than that occurs for plain SLNs.

As for the particle size, in TEM images (Figure 1d–f), many large particles (up to 1000 nm) are present, and the question of how to explain the differences between them and the sizes analyzed by PCS can be raised. Figure 1d–f show particles with variable sizes that are representative of the distribution of the sample. To confirm this observation, we analyzed the TEM images and estimated the size distribution of 100 particles by means of ImageJ Software 1.8.0. (GitHub, Inc., San Francisco, CA, USA). The average size is equal to 265 ± 98 nm, with the smallest particle size of 83 nm and the largest one a diameter of 617 nm. This analysis evinces the broad distribution of the sample on one hand and that the most abundant particles display an average size close to that obtained by PCS measures. Noteworthy, by looking at the Gaussian distribution of Figure 1d, it can be observed that the base of curve is quite broad, with the smallest particles below 100 nm and the largest close to 1 µm. Therefore, we can conclude that the two types of measurements are not conflicting each other and reflect the real size distribution of the sample.

In addition to these findings, the results of the solid-state studies on CIT-SLNs, taken together, suggest that these nanocarriers are characterized by a most reduced crystallinity of the drug and an amorphous internal structure of these solid particles. Moreover, the CIT-SLNs at solid state maintain features of the solid lipid matrix and the plain SLNs. It induced us to likely believe that CIT-SLNs possess good mucoadhesive properties as already demonstrated for plain SLNs [43]. An interesting outcome observed during these solid state studies was the mentioned presence of two endothermic peaks at 130 °C and 150 °C before the melting of the drug in the DSC thermogram of CIT-SLNs and we have already pointed out that more detailed investigation in this regard is necessary to draw definitive conclusions. On the whole, our hypothesis is that CIT may exist at solid state in two polymorphic crystalline forms and, moreover, that it melts with decomposition at temperatures >250 °C.

As for the physical stability of CIT-SLNs on storage, the results confirm that these nanocarriers possess essentially low stability, in agreement with that suggested by the essentially neutral zeta potential value. From the results reported in Figure 4a,b it can be deduced that the best storage conditions occur for few days at the lowest temperature (i.e., 4 °C) and for one day at most at the highest temperature investigated (i.e., 37 °C).

Concerning biological evaluation, both undifferentiated and differentiated 6-OHDA-treated SH-SY5Y cells have been considered as suitable accepted experimental models for in vitro studies on PD [9,53] (PMID: 10936180). However, as the main objective of this work was to evaluate if CIT encapsulation within SLNs may improve the drug efficacy in the prevention of 6-OHDA toxicity, we started from the reported experience of two papers [9,54] in which the 6-OHDA induced cytotoxicity in undifferentiated SH-SY5Y cells has been described.

The results of the cells viability test on SH-SY5Y cells (Figure 6) demonstrate how CIT, encapsulated within SLNs, is able to remove the cellular damage induced by 6-OHDA, a neurotoxin already used to set up experimental models of PD [55]. In our experimental conditions cytotoxicity induced by the neurotoxin 6-OHDA was not severe but still informative about alterations of degenerative pathways caused by different treatments. Notably, CIT encapsulation in SLNs results in a greater cytoprotective efficacy than free CIT, in order to restore cell viability [9,56]. Additionally, it seems that the protective effect of CIT-SLNs is not related to the lipid matrix of the carrier, since plain SLNs, by themselves, do not counteract the toxic effect of 6-OHDA (Figure 6). In details, from Figure 6 it could be derived that 6-OHDA treatment achieved almost 100% of cell viability and cytoprotection due to CIT-SLNs could not be evident. However, it may be argued that especially in the initial stages of neurological pathologies such as PD, the degenerative stimuli are not necessarily so severe as to cause cell death. Moreover, to shed light on molecular mechanisms of degeneration it is important that the toxic pathways can be modulated, a too strong stimulus may ultimately prove itself to be uninformative. In our experimental conditions, the 40 μM 6-OHDA concentration is statistically significant and produces a decrease of about one fifth with respect to the cell viability of the untreated cell culture (Figure 6). It is a dynamic process and after 48 h the toxic efficacy of 6-OHDA dose is almost complete. Upon these conditions, which could be equated to an early stage of pathological degeneration, the effect of CIT-SLNs was proved to be significantly different from the effect of free CIT, suggesting that there is probably a molecular mechanism underlying this finding which merits further research.

In particular, in the literature, comparative studies between the effects of free antioxidant agents and their corresponding nanocarriers evidenced the higher impact of nanoencapsulation more than the free agents. For instance, Picone et al. found that the encapsulated antioxidant ferulic acid was more active than free form once cells were treated with recombinant Aβ as a neurotoxin [57]. Furthermore, another antioxidant, L-carnitine, was shown to be more effective in comparison to the free form in protecting HUVEC cells from oxidative stress induced by oxygen dioxide [58]. Likewise, resveratrol and etoposide have been proved to be more cytotoxic in the form of encapsulated SLNs than in the free ones as anti-cancer agents [59,60]. For a better understanding of CIT (and CIT-SLNs) cellular effect, DAPI staining method was adopted (Figure 7). It is widely used for visualization of nuclear changes during apoptosis or necrosis such as the formation of nuclear fragmented bodies, condensed or deformed nuclei [61]. In fact, even if the cell death induced by the experimental condition used in our study is not widespread, CIT-SLNs decrease the nuclear condensation detectable in the treated cells restoring, at least qualitatively, the nuclear morphology present in the controls. Definitely, these effects are markedly more evident with CIT-SLNs than free CIT (Figure 7). To understand why CIT-SLNs are more effective as neuro-protective systems than free CIT maybe it should be pointed out as follows. Although showing different mechanisms, the previously mentioned ferulic acid, etoposide, resveratrol are substances exhibiting instability in aqueous solution as well as CIT, so, according to Refs. [56,62], it could be deduced that CIT introduction into a protected matrix (such as in the SLNs) could improve its physical stability until release or possible fusion of the SLNs with the cell membrane. Then, the consequent benefit of SLNs internalization could be CIT release directly into the cytoplasm of SH-SY5Y cells, as confocal images of the cellular uptake of the fluorescent dye FITC seems to suggest (Figure 5), once FITC-SLNs replaced unlabeled CIT-SLNs. Remarkably, FITC shows a high efficiency of energy transfer from absorbed to emitted ultraviolet light and is widely used as a tracer in cell cytoplasm up-take assays [42,63,64,65]. Furthermore, the benefit connected to FITC is that it is not leaked from SLNs until 24 h from the administration as already reported [66].

On the base of the results from Figure 5, our hypothesis is that one possible reason to clarify at present time why CIT-SLNs performed better than pure CIT is that CIT-SLNs help to increase the intracellular levels of CIT. Overall, to confirm such a hypothesis, further investigations will be needed to shed light on the matter definitively.

Previous investigations pointed out the interesting performance of CIT in the treatment of several neurological diseases, the anti-oxidative stress related effects of CIT, and the maintenance of biomembrane integrity [62]. In this context, a better understanding of the processes allowing CIT to protect neuronal survival from oxidative stress may provide new working hypotheses about the mechanism of action of this biologically active and safe substance. In this sense, Travaglione et al. recently suggested that the bacterial toxin CNF1 protects SH-SY5Y cells against 6-OHDA-induced cell damage through a mechanism which involved mitochondrial functionality, autophagy and the redox profile of the cells [54]. These findings prompt us to verify in a future step the appropriateness of this hypothesis, investigating on the possible link occurring among oxidative stress, functional status of mitochondria and autophagy in differentiated neuron-like SH-SY5Y cells. For instance, morphological and/or biochemical/molecular pathways related to 6-OHDA induced cell suffering and death (and the relative capability of encapsulated CIT to prevent/counteract this condition) may be investigated in differentiated neuron-like SH-SY5Y cells. It is well known that “autophagy, oxidative stress, and endoplasmic reticulum stress are assumed to act as a “double-edged sword” for neurodegenerative conditions, due to their neurodegenerative-suppressing and/or neurodegenerative promoting actions [67].

## 5. Conclusions

In conclusion, in the present paper, we describe an advantageous strategy for encapsulating CIT into SLNs based on a self-emulsifying lipid as Gelucire^®^ 50/13 in order to gain information on therapeutic CIT use in parkinsonian syndrome. Our biological evidence showed an enhanced protective effect of CIT-SLNs on human neuroblastoma SH-SY5Y cells exposed to a mild oxidative stress induced by 6-OHDA treatment, in comparison with free CIT. Taking into account some recent hints of literature, this anti-oxidative stress of CIT-SLNs may occur by triggering autophagy activity and mitochondrial fusion. Hence, to elucidate the therapeutic role of CIT in parkinsonian syndrome, it seems necessary to investigate such aspects, and it is mandatory that future research address these directions. The use of differentiated SH-SY5Y cells could be of interest in a future following step of the research, in which morphological and/or biochemical/molecular pathways related to 6-OHDA induced cell suffering and death (and the relative capability of the encapsulated CIT to prevent/counteract this condition) may be deeply investigated.

## Figures and Tables

**Figure 1 pharmaceutics-14-01827-f001:**
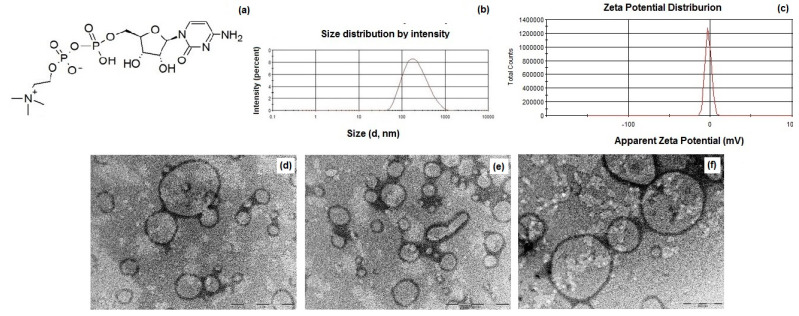
(**a**) Chemical structure of citicoline (CIT); (**b**) particle size distribution of CIT-SLNs; (**c**) zeta potential distribution of CIT-SLNs; (**d**–**f**) TEM images of CIT-SLNs [the scale bar of panels (**d**) and (**e**) is equal to 1 µm, while in (**f**) corresponds to 500 nm]. (For further details on TEM images and particle size, see Appendix A).

**Figure 2 pharmaceutics-14-01827-f002:**
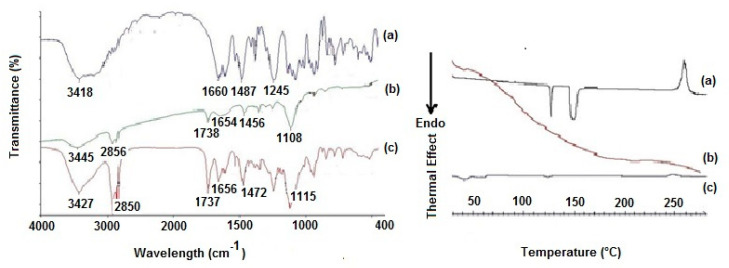
Left panel: FT-IR spectra of (**a**) pure CIT, (**b**) plain SLNs and (**c**) CIT-SLNs. Right panel: DSC profiles of (**a**) pure CIT, (**b**) plain SLNs and (**c**) CIT-SLNs.

**Figure 3 pharmaceutics-14-01827-f003:**
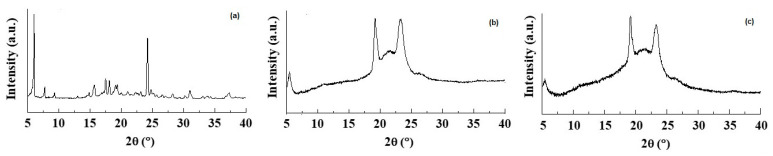
X-ray diffraction patterns of (**a**) pure CIT, (**b**) plain SLNs and (**c**) CIT-SLNs.

**Figure 4 pharmaceutics-14-01827-f004:**
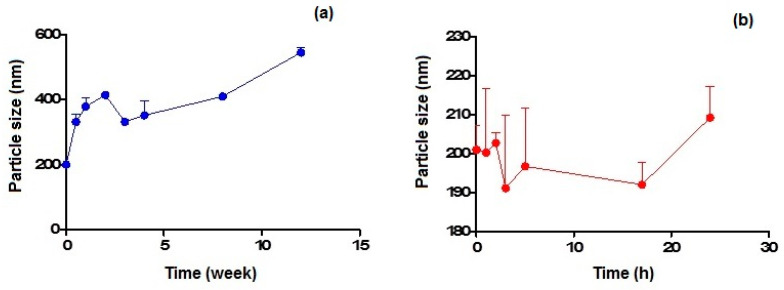
Particle size variation of CIT-SLNs after incubation at: (**a**) 4 °C and (**b**) 37 °C.

**Figure 5 pharmaceutics-14-01827-f005:**
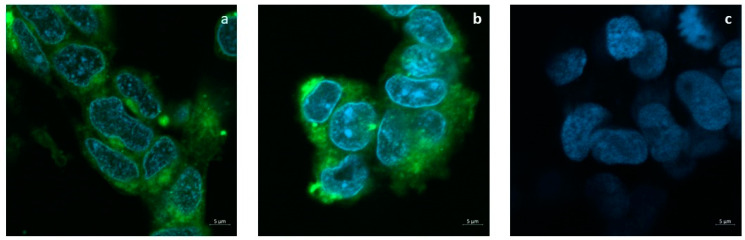
Uptake assay of FITC-loaded SLNs. Confocal microscopy imaging shows that FITC molecules encapsulated into SLNs penetrate the cytoplasm of SH-SY5Ycell cultures. (**a**) FITC-loaded SLNs (50 µg/mL); (**b**) FITC-loaded SLNs (100 µg/mL); (**c**) Plain SLNs (50 µM).

**Figure 6 pharmaceutics-14-01827-f006:**
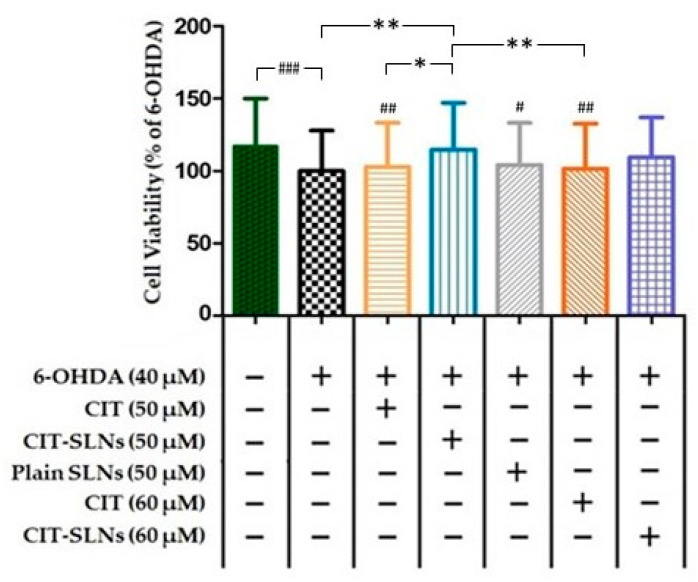
Measurement of percentage of SH-SY5Y viable cells by WST-8/CCK8 assay. The results are expressed as the mean ± standard deviation; *n* = 5, * *p* < 0.05, ** *p* < 0.01, # *p* < 0.05 vs. no-treatment control, ## *p* < 0.01 vs. no-treatment control, ### *p* < 0.001 vs. no-treatment control. (See Appendix A for the original raw data).

**Figure 7 pharmaceutics-14-01827-f007:**
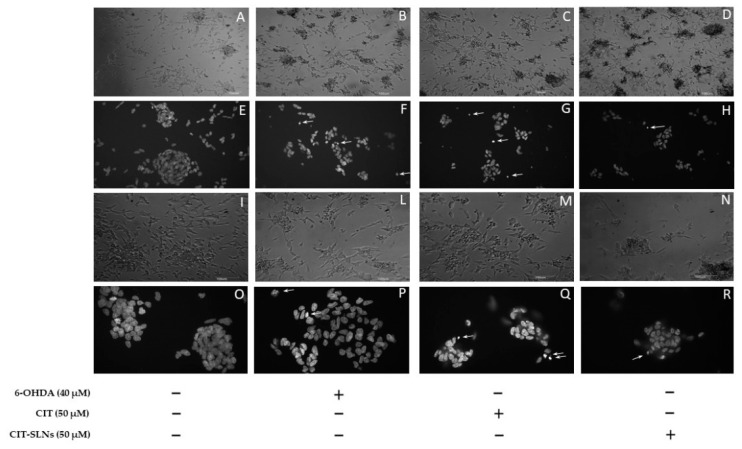
Morphological effects of CIT-SLN treatment. SH-SY5Y cells were treated with 6-OHDA (40  μM) in the presence or absence of a pre-treatment (2 h) with free CIT or CIT-SLNs (50  μM each one) and incubated for 24  h. Phase-contrast micrographs (**A**–**D**,**I**,**L**–**N**) or images of DAPI staining (**E**–**H**,**O**–**R**). Phase-contrast micrographs (first and third row) or images of DAPI staining (second and fourth row). Arrows point to nuclear fragmentation. Micrographs are representative of three different experiments. Magnification: 6,3× (**A**–**D**); 10× (**I**,**L**–**N**); 20× (**E**–**H**); 40× (**O**–**R**).

**Table 1 pharmaceutics-14-01827-t001:** Physicochemical characterization of prepared SLNs. All values are expressed as mean ± standard deviation of at least eight replicates (*n* = 8).

Formulation	Size (nm)	PDI ^a^	Zeta Potential (mV)	E.E. CIT (%)
Plain SLNs	141 ± 11	0.34 ± 0.08	−9.2 ± 0.7	-
CIT-SLNs	201 ± 24	0.45 ± 0.08	−2.2 ± 0.2	80 ± 7

^a^ PDI: Polydispersity index.

## Data Availability

Not applicable.

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
