# Peer review of "The Encapsulation of Citicoline within Solid Lipid Nanoparticles Enhances Its Capability to Counteract the 6-Hydroxydopamine-Induced Cytotoxicity in Human Neuroblastoma SH-SY5Y Cells"

_pharmaceutics, 2022, doi:10.3390/pharmaceutics14091827_

Round 1

Reviewer 1 Report (Previous Reviewer 1)

Can be accepted 

Author Response

We are grateful to the Reviewer for his/her decision

Reviewer 2 Report (Previous Reviewer 4)

The authors responded to the comments and revised the manuscript.

Author Response

We are grateful to the Reviewer for this comment

Reviewer 3 Report (Previous Reviewer 2)

The manuscript was resubmitted after a revision. Although it was improved, there are still some shortcomings, as follows.

1. The preparation of CIT-SLNs: there was only one CIT-SLNs formulation. In other words, there was no screening or optimization of the formulation. As a result, the PDI values were high (0.34 - 0.45). The formulation needs more optimization to have a better size distribution before further steps.

2. In Figure 6, many significant symbols were used. However, it seemed that there was no significance among treatments. The authors should carefully check the data and perform the statistical test. The authors can provide raw data in a supplementary file to confirm these statistical results were correct.

3. Preparation of Citicoline (CIT) loaded Gelucire® 50/13 based-SLNs (CIT-SLNs): why was CIT added to the water phase but not the lipid phase? Since CIT is hydrophilic and was added to the water phase during the preparation of CIT-SLNs, the EE of 80% (Table 1) seemed to be unreasonably high. How can the authors explain it?

4. The release data were removed; why? There were some concerns regarding these data in the previous version (it is strange that the cumulative CIT released from CIT-SLNs decreased to ~15% at 48 h, about 2-fold lower than the pure drug. The authors should explain: (i) why the cumulative released CIT decreased and (ii) why the decrease in CIT-SLNs was greater than that in pure CIT).

5. TEM images were unclear. Many large particles (up to 1000 nm) are present. How to explain the differences between them and the sizes analyzed by PCS.

Author Response

The manuscript was resubmitted after a revision. Although it was improved, there are still some shortcomings, as follows.

Q1. The preparation of CIT-SLNs: there was only one CIT-SLNs formulation. In other words, there was no screening or optimization of the formulation. As a result, the PDI values were high (0.34 - 0.45). The formulation needs more optimization to have a better size distribution before further steps.

R1. The authors would like to acknowledge the Reviewer for this issue. We are aware that optimization of CIT-SLN formulation could be of interest but, in our opinion, this is out of the scope of the present work. However, we are considering to address the issue in a forthcoming research work.

Q2. In Figure 6, many significant symbols were used. However, it seemed that there was no significance among treatments. The authors should carefully check the data and perform the statistical test. The authors can provide raw data in a supplementary file to confirm these statistical results were correct.

R2. As suggested by the Reviewer, an Excel data sheet containing raw data was provided in “Supplementary Materials”, where the statistical approach described in Section 2.8 of the revised manuscript can be verified. In the edited caption to Figure 6 of the revised manuscript we have also indicated to the reader to access to Supplementary Materials Section

Q3. Preparation of Citicoline (CIT) loaded Gelucire® 50/13 based-SLNs (CIT-SLNs): why was CIT" added to the water phase but not the lipid phase? Since CIT is hydrophilic and was added to the water phase during the preparation of CIT-SLNs, the EE of 80% (Table 1) seemed to be unreasonably high. How can the authors explain it?

R3. In the Discussion Section, we have cited our previous works where in the presence of hydrophilic drug substances high E.E. percentages have been achieved (Refs. 41-43,46). Furthermore, we have better clarified that the chemical affinity is involved in the decision of disperding CIT in the aqueous phase of the emulsion leading to the final SLNs, rather than the co-melting approach with the lipid Gelucire® 50/13 which was followed for hydrophobic drug substance loading (Ref. 47).

Q4. The release data were removed; why? There were some concerns regarding these data in the previous version (it is strange that the cumulative CIT released from CIT-SLNs decreased to ~15% at 48 h, about 2-fold lower than the pure drug. The authors should explain: (i) why the cumulative released CIT decreased and (ii) why the decrease in CIT-SLNs was greater than that in pure CIT).

R4. On August 3, 2022, the authors received from the Academic Editor the advise before peer review: “Being CIT unstable in the release medium, the release study is hard to interpret. Therefore, it should be removed”. All the authors have been in agreement with this suggestion, and, therefore, results and discussion concerning in vitro studies have been deleted from the original version

Q5. TEM images were unclear. Many large particles (up to 1000 nm) are present. How to explain the differences between them and the sizes analyzed by PCS.

R5. In the images included into the manuscript many large particles are visible and they are not fully representative of the distribution of the sample.

Hereafter, we provide other TEM images in Supplementary Materials that contain many small particles and denote the broad distribution of the SLNs size and the coexistence of particles with sizes ranging from 100 to 1000 nm.

Indeed, the analysis of the TEM size of more than 100 particles by Image J Software (See Supplementary Materials) evidenced such broad distribution. The SLNs have an average size equal to 277 ± 198 nm, being the smaller particles close to 70 nm while the largest one to 980 nm.

A similar broad distribution can be observed after PCS analysis where the data of Table 1 indicate that the CIT-SLNs have an average size of 201 nm but a large PDI equal to 0.45. In addition, by looking at the Gaussian distribution of Figure 1c, it can be observed that the base of curve is quite broad with the smaller particles below 100 nm, and the largest close to 1 µm.

Therefore, we can conclude that the two types of measures are not conflicting and reflect the real size distribution of the sample

Round 2

Reviewer 3 Report (Previous Reviewer 2)

The authors appropriately modified the manuscript and responded to all previous comments. The only remaining concern is the significance of the data presented in Figure 6. The authors provided the raw data with statistical results. This reviewer is not an expert in statistics and cannot evaluate those data. However, looking at similar means among groups and relatively high SDs in each group, it is hard to believe that there are significant differences. Thus, this reviewer strongly recommends the authors get help from an expert in statistics to avoid incorrect data before publishing the manuscript.

Author Response

Q1. The authors appropriately modified the manuscript and responded to all previous comments.

R1. We are grateful to the Reviewer to appreciate our work

Q2. The only remaining concern is the significance of the data presented in Figure 6. The authors provided the raw data with statistical results. This reviewer is not an expert in statistics and cannot evaluate those data. However, looking at similar means among groups and relatively high SDs in each group, it is hard to believe that there are significant differences. Thus, this reviewer strongly recommends the authors get help from an expert in statistics to avoid incorrect data before publishing the manuscript.

R2. Concerning this issue, the coefficients of variation CV (i.e.,  the ratio between the standard deviation and the mean) of the examined runs shown in the Excel file of Supplementary Materials are in the range 0.278-0.305 (see also Excel file enclosed to this message). The higher the CV, the higher the standard deviation relative to the mean. In general, a CV value greater than 1 is often considered high. Hence, the CV values and, consequently, the corresponding standard deviations herein reported can be considered not high but, conversely, reasonable, particularly taking into account that they are associated to biological data.

This manuscript is a resubmission of an earlier submission. The following is a list of the peer review reports and author responses from that submission.

Round 1

Reviewer 1 Report

Authors reported the Encapsulation of Citicoline within Solid Lipid Nanoparticles en- 2 hances its capability to counteract the 6-hydroxydopamine-in- 3 duced cytotoxicity in human neuroblastoma SH-SY5Y cells.

my recommendations are following

Abstract:

Abstract are very confusing. No any correlation between the sentence. Authors are advised to rewrite tie abstracts in scientific manners.

Introduction

Authors are advised to write some recent reported literature related to recent paper. How his research will be more better than recently reported work??

Results:

zeta potential of the formulation is not within the optimum range. How authors will say that his formulation is stable??

Plz add zeta size and zeta potential figure (if not possible then add as a supplementary file.

Reviewer 2 Report

Review comments on pharmaceutics-1787512: Encapsulation of Citicoline within Solid Lipid Nanoparticles enhances its capability to counteract the 6-hydroxydopamine-induced cytotoxicity in human neuroblastoma SH-SY5Y cells

The manuscript by Margari et al. described the preparation of Citicoline-Loaded Solid Lipid Nanoparticles (CIT-SLNs) and in vitro evaluations of the formulation. The study included 3 parts: preparation of CIT-SLNs, characterization of CIT-SLNs, and CIT-SLNs cytotoxicity in SH-SY5Y cells. There are several concerns regarding each part as follows. Considering the below comments, this manuscript is unsuitable for publication in Pharmaceutics in the current form.

1. The preparation of CIT-SLNs: there was only one CIT-SLNs formulation. In other words, there was no screening or optimization of the formulation. As a result, the PDI values were high (0.28-0.40 for plain SLNs and 0.39-0.51 for CIT-SLNs). The formulation needs more optimization to have a better size distribution before further steps.

2. Evaluation of CIT-SLNs on SH-SY5Y cells: the cytotoxicity was insufficient to gain more insight into the effects of CIT-SLNs on SH-SY5Y cells. In addition, from the data shown in Figure 5, it seemed that there was no significance among treatments, as indicated by many significant symbols. Therefore, the authors should carefully check the data and perform the statistical test. For example, data in columns #3 and #4 were marked with *** (p<0.001), but it seemed impossible.

3. The study aimed to prepare CIT-SLNs and evaluate its effects on SH-SY5Y cells compared with pure CIT and plain SLNs. However, in the results and discussion, the role of the formulation was not clarified. There was only discussion based on the performance of CIT. It is required to clarify why CIT-SLNs performed better than pure CIT. One possible reason is that CIT-SLNs help to increase the intracellular level of CIT.

4. Section 2.2- Quantitative Determination of Citicoline: when using a UV-Vis method to quantify a compound in a formulation, the method should be validated in terms of specificity. Some components in the formulation can have absorption at the selected UV-Vis wavelength and affect the accuracy of the targeted compound. The authors should include data to clarify it.

5. Section 2.3- Preparation of Citicoline (CIT) loaded Gelucire® 50/13 based-SLNs (CIT-SLNs): why was CIT added to the water phase but not the lipid phase? Since CIT is hydrophilic and was added to the water phase during the preparation of CIT-SLNs, the EE of 80% (Table 1) seemed to be unreasonably high. How can the authors explain it?

6. FTIR, DSC, and XRD: were the plain SLNs and CIT-SLNs in suspension or freeze-dried form?

7. Lines 115-117: the authors should cite relevant references.

8. Lines 146-148: the authors should include details of the freeze-drying. Were cryoprotectants used?

9. In TEM images (Figures 1c and 1e), many large particles (up to 1000 nm) are present. How to explain the differences between them and the sizes analyzed by PCS.

10. PDI values should be presented as mean ± standard deviation.

11. Figure 4c: it is strange that the cumulative CIT released from CIT-SLNs decreased to ~15% at 48 h, about 2-fold lower than the pure drug. The authors should explain: (i) why the cumulative released CIT decreased and (ii) why the decrease in CIT-SLNs was greater than that in pure CIT.

12. Quality of Figure 2 should be improved. Figures 4a and 4b should be converted to line graphs.

13. What can be concluded from the FTIR data? There should be a discussion on them.

14. Lines 496-498: did this study involve animals?

15. Many typos and grammar errors need to be carefully checked and corrected (e.g., lines 93-95, 108-110, 140, etc.).

Reviewer 3 Report

The paper entitled Encapsulation of citicoline within solid-lipid nanoparticles enhances its capability to counteract the 6-hydroxydopamine-induced cytotoxicity in human neuroblasma SH-5Y5Y cells” describes very well the synthesis strategy to develop a nasal formulation based on solid lipid nanoparticles in order to enhance the therapeutic use of citocoline in the Parkinson disease treatment.

The paper is well-structured with good presentation. Most of literature is from recent years indicating that the authors have been guided by the latest studies.

I have a small question: How can the authors explain the increase of diameter size from 141 nm (plain SLNs) to 201 nm (CIT-SLNs)?

In conclusion, this paper is well-written and might be interesting and useful for researchers in this field and the manuscript can be accepted for publication in Pharmaceutics journal.

Reviewer 4 Report

In this manuscript, Andrea Margari and his colleagues described the development of an advantageous strategy to enhance the therapeutic use of CIT (Citicoline) in Parkinson’s disease (PD). The encapsulated CIT in SLNs (Solid Lipid Nanoparticles) showed an increased protective effect on the human neuroblastoma SH-SY5Y cells compared to the free CIT. Oxidative stress has been induced by 6-OHDA (6-hydroxydopamine) treatment to mimic PD, which is a widely accepted model of DAergic neurons for PD research. Because there are no drugs for modifying the course of PD, the investigation of new therapeutic agents is important and needed.

The manuscript is well written, in a fluent and clear style. The structure of the introduction is logical and extensive, the discussion is correct, but I have some remarks:

There is a long part about the intranasal administration of encapsulated therapeutic agents in the introduction. The encapsulated CIT in lipid-based nanocarriers was prepared for the purpose of a nasal administration to bypass the blood-brain barrier. The authors investigated the in vitro release kinetics from CIT-SLNs in an artificial nasal fluid mixture, but it would have been good to see the effect of CIT-SLNs in an in vivo study or at least the permeability capability of CIT-SLNs through a model of the nasal epithelium RPMI 2650 cell line. There are several accessible sources in the literature, for instance: PMID: 19007885, PMID: 27702697. This deficiency is a limitation of this study and it would be required to mention in the discussion part. Furthermore, the completion of these results with an extra experiment should be considered.

Minor concern:

For the biological evaluations, the human neuroblastoma SH-SY5Y cell line was used. Both the differentiated and undifferentiated SH-SY5Y cell lines are accepted as a model of DAergic neurons.  Some differentiated SH-SY5Y cells, for instance, with retinoid acid (RA) are more suitable models because differentiation makes SH-SY5Y cells more analogous to DAergic neurons, which is a reasonable model for the exploration of the pathogenesis of PD and the evaluation of therapies (PMID: 10936180). To compare the effect of CIT-SLNs on a differentiated and undifferentiated cell culture would be essential and more informative. Because of this, a new cell viability assay on differentiated cell culture should be performed.

Comments:

- the abbreviation of Citicolin is unnecessary in the subtitle of Material and Methods (2.3.)

- the manuscript has to be checked for missing/extra spaces in the vicinity of metric units and numbers: ”°C” and “h” (for ex.: there are extra spaces in line 213, but not in line 230-231)

- there are extra dots in lines 94 and 140

- the metric units used are not consistent: “hours or h” and “× or X”

- please add the (commercial) source of all reagents, plasticware and instruments, it should be written uniformly in all cases: incomplete in lines 137, 145, 158, 177, 215, 217, 225, 236

Reviewer 5 Report

The manuscript entitled "Encapsulation of Citicoline within Solid Lipid Nanoparticles enhances its capability to counteract the 6-hydroxydopamine-induced cytotoxicity in human neuroblastoma SH-SY5Y cells" reports the production and effect of SLN entrapping citicoline in a cell model of Parkinson. Overall, the results are well reported and discussed. Only a few details should be clarified:

2.1. Materials - Please, list all the reagents used, including cell lines, cell medium components, etc.

3.4.1. - Why did the authors use a 6-OHDA concentration that only causes a 17.1% of the decrease in cell viability? Have you tested higher concentrations of 6-OHDA? Looking at the graphic (Figure 5), 6-OHDA treatment achieved almost 100% of cell viability. How can the authors correctly see a protective effect of CIT-loaded SLNs in a situation where the cytotoxicity was not severe at all? Although authors have observed morphological changes in cell culture using 6-OHDA (40 uM), this seems not to influence cell viability.

Figure 6 - Please, verify the legend below the micrographs. According to the text, Figures 6H and 6R correspond to the pre-treatment of cells with e CIT-SLNs before 6-OHDA treatment. However, below this column, the legend states that all the column reports the data obtained with 6-OHDA (-). 

For these reasons, I recommend minor revisions to this manuscript.